# Proteomic Profiling Differentiates Lymphoma Patients with and without Concurrent Myeloproliferative Neoplasia

**DOI:** 10.3390/cancers13215526

**Published:** 2021-11-03

**Authors:** Johanne Marie Holst, Marie Beck Enemark, Martin Bjerregaard Pedersen, Kristina Lystlund Lauridsen, Trine Engelbrecht Hybel, Michael Roost Clausen, Henrik Frederiksen, Michael Boe Møller, Peter Nørgaard, Trine Lindhardt Plesner, Stephen Jacques Hamilton-Dutoit, Francesco d’Amore, Bent Honoré, Maja Ludvigsen

**Affiliations:** 1Department of Hematology, Aarhus University Hospital, 8200 Aarhus, Denmark; johahols@rm.dk (J.M.H.); mariem@rm.dk (M.B.E.); martipde@rm.dk (M.B.P.); trihyb@rm.dk (T.E.H.); frandamo@rm.dk (F.d.); 2Department of Clinical Medicine, Aarhus University, 8000 Aarhus, Denmark; stephami@rm.dk; 3Department of Pathology, Aarhus University Hospital, 8200 Aarhus, Denmark; krislaur@rm.dk; 4Department of Hematology, Vejle Hospital, 7100 Vejle, Denmark; Michael.Roost.Clausen@rsyd.dk; 5Department of Hematology, Odense University Hospital, 5000 Odense, Denmark; henrik.frederiksen@rsyd.dk; 6Department of Pathology, Odense University Hospital, 5000 Odense, Denmark; Michael.Boe.Moeller@rsyd.dk; 7Department of Pathology, Herlev Hospital, 2730 Herlev, Denmark; Peter.H.Noergaard@regionh.dk; 8Department of Pathology, Rigshospitalet, 2100 Copenhagen, Denmark; trine.lindhardt.plesner@regionh.dk; 9Department of Biomedicine, Aarhus University, 8000 Aarhus, Denmark; bh@biomed.au.dk

**Keywords:** angioimmunoblastic T-cell lymphoma (AITL), diffuse large B-cell lymphoma (DLBCL), myeloproliferative neoplasia (MPN), proteomics

## Abstract

**Simple Summary:**

Patients are diagnosed with myeloproliferative neoplasia (MPN) and lymphoma more frequently in the population than expected, which has led to the hypothesis that the two malignancies may, in some cases, be pathogenetically related. In this study, lymphoma patients with and without MPN show subtle but important differences in the protein expression that enables the clustering of the lymphomas, thus indicating the differences at the molecular level between the lymphoma malignancies with and without MPN, and strengthening the hypothesis that the lymphoma and MPN may be biologically related.

**Abstract:**

Myeloproliferative neoplasia (MPN) and lymphoma are regarded as distinct diseases with different pathogeneses. However, patients that are diagnosed with both malignancies occur more frequently in the population than expected. This has led to the hypothesis that the two malignancies may, in some cases, be pathogenetically related. Using a mass spectrometry-based proteomic approach, we show that pre-treatment lymphoma samples from patients with both MPN and lymphoma, either angioimmunoblastic T-cell lymphoma (MPN-AITL) or diffuse large B-cell lymphoma (MPN-DLBCL), show differences in protein expression compared with reference AITL or DLBCL samples from patients without MPN. A distinct clustering of samples from patients with and without MPN was evident for both AITL and DLBCL. Regarding MPN-AITL, a pathway analysis revealed disturbances of cellular respiration as well as oxidative metabolism, and an immunohistochemical evaluation further demonstrated the differential expression of citrate synthase and *DNAJA2* protein (*p* = 0.007 and *p* = 0.015). Interestingly, *IDH2* protein also showed differential expression in the MPN-AITL patients, which contributes to the growing evidence of this protein’s role in both myeloid neoplasia and AITL. In MPN-DLBCL, the disturbed pathways included a significant downregulation of protein synthesis as well as a perturbation of signal transduction. These results imply an underlying disturbance of tumor molecular biology, and in turn an alternative pathogenesis for tumors in these patients with both myeloid and lymphoid malignancies.

## 1. Introduction

Myeloproliferative neoplasia (MPN) and lymphoma are traditionally thought to develop by different pathogenetic mechanisms, leading to the occurrence of distinct diseases [1,2]. Lymphomas are malignancies that are derived from mature lymphocytes whereas MPN are clonal hematopoietic stem cell disorders that are characterized by the malignant proliferation of one or more of the myeloid-derived cell lineages [1,2]. Interestingly, patients that are diagnosed with both malignancies, i.e., MPN and lymphoma, occur at higher frequencies than expected compared with the background population [3,4,5]. Such observations have fostered the hypothesis that these two hematological malignancies, in some cases, may share molecular alterations representing a possible pathogenetic relationship. If so, the tumor samples from patients harboring both diseases may show differences in biology compared with the tumor samples from patients with sporadic lymphoma and no accompanying myeloproliferative disease. MPN encompasses the entities essential thrombocythemia (ET), polycythemia vera (PV), primary myelofibrosis (PMF), chronic myeloid leukemia, and MPN-unclassifiable (MPN-U) [2,6]. In spite of their distinct clinicopathological features, MPN can be viewed as a disease spectrum with shared genetic and clinical aberrations [7]. Well-known recurrent alterations include mutations involving the *JAK2*, *CALR* or *MPL* genes, which are found in approximately 90% of patients with MPN. Moreover, patients with MPN are, in general, at an increased risk of subsequent progression to acute myeloid leukemia [7].

Lymphomas encompass a wide range of lymphocyte-derived malignancies, including diffuse large B-cell lymphoma (DLBCL) and angioimmunoblastic T-cell lymphoma (AITL), both of which are aggressive lymphoid neoplasia. DLBCL, the most frequent type of non-Hodgkin lymphoma, is derived from mature B cells of either the germinal center type or post-germinal center type, which is a dichotomy that reflects an underlying genomic heterogeneity [8,9,10]. AITL is one of the more common subtypes of T-cell lymphoma, originating from mature T follicular helper cells [11,12]. AITL develops via a multistep oncogenic pathway involving frequent *RHOA* mutations as well as recurrent mutations in the epigenetic modifier genes, e.g., *TET2*, *DNMT3A*, and *IDH2*. These modulator genes are also commonly found to be mutated in myeloid neoplasms, including MPN [12,13,14,15,16,17]. Various factors may act at different levels to influence the final properties of the encoded proteins, and the discrepancies between gene expression and the final functional protein phenotype may, for example, be caused by translational inhibition/activation or by post-translational modifications [18].

In a recent retrospective cohort study, we reported inferior survival in patients with both MPN and DLBCL (MPN-DLBCL patients) compared with a matched DLBCL control group without MPN [6]. We found no significant difference in survival for patients with MPN and AITL (MPN-AITL patients) compared with an AITL control group, although this analysis was limited by a small sample size [6]. Nonetheless, our findings suggested that further investigations comparing the tumor biology of lymphomas that develop in patients both with and without concomitant MPN might be of interest. In the present study, we used mass spectrometry-based proteomics to investigate if we could identify differentially expressed proteins in pretreatment lymphoma samples from (i) MPN-AITL patients and (ii) MPN-DLBCL patients, compared with control groups of sporadically occurring lymphomas.

## 2. Patients and Methods

### 2.1. Patient Samples

Analyses were performed on formalin fixed, paraffin-embedded (FFPE) primary diagnostic lymphoma specimens from 34 patients. These included tumor tissue from patients with concurrent diagnoses of either MPN and AITL (MPN-AITL, *n* = 8) or of MPN and DLBCL (MPN-DLBCL, *n* = 9) compared with reference AITL (R-AITL, *n* = 8) and reference DLBCL (R-DLBCL, *n* = 9), respectively. The MPN-AITL and MPN-DLBCL patient samples included in the present study originate from a Danish cohort of patients that were diagnosed with both MPN and lymphoma between 1990–2015 and have previously been described [6]. The patients with reference lymphomas were identified through the Danish Lymphoma Registry (LYFO) [19] and matched according to the type of tissue and age at diagnosis. Biopsies were reviewed by an expert hematopathologist, diagnoses being confirmed, based on the 2017 revision of the WHO Classification of Tumours of Haematopoietic and Lymphoid Tissues [1]. Patient clinical data was obtained from LYFO [19] and from the Danish National Chronic Myeloid Neoplasia Registry [20]. To exclude cases of secondary and therapy-related MPN, the cohort was based on patients that were diagnosed either with both diseases simultaneously (i.e., diagnosed no more than six months apart) or with MPN first and subsequently with lymphoma [6]. Of the eight MPN-AITL patients, data on *IDH2* gene mutations were available in five samples [6]. Of these, two harboured an R172M missense mutation and one harboured two different missense mutations (R172G and R140Q).

### 2.2. Identification of Differentially Expressed Proteins

In order to identify the differentially expressed proteins between tumors from lymphoma patients with and without MPN, a label-free quantification nano liquid chromatography-tandem mass spectrometry (LFQ nLC-MS/MS)-based proteomic analysis was performed. The procedure is described in detail in the Supplementary methods. In brief, proteins were extracted from FFPE lymphoma tissues [21]. Extracted proteins were proteolytically digested into peptides, which were then separated by nano liquid chromatography and analysed in the mass spectrometer (Orbitrap Fusion, Thermo Fisher Scientific, Waltham, MA, USA) [21]. The identified peptides were used to search the human protein database from UniProt Consortium (Uniprot Knowledgebase, The Universial protein Resource, www.uniprot.org, database downloaded 10 April 2018) using MaxQuant v1.5.5.1 [22] (Max Plank Institute of Biochemistry, Martinsried, Munic, Germany) and Perseus v1.6.2.3 [23] Max Plank Institute of Biochemistry, Martinsried, Munic, Germany) in order to identify the protein composition within each sample [21]. Bioinformatic analysis was performed by using Ingenuity Pathway Analysis (QIAGEN Inc., Hilden, Germany, https://www.qiagenbioinformatics.com/products/ingenuity-pathway-analysis, 7 June 2020) with developed algorithms described by Krämer [24].

### 2.3. Immunohistochemical Evaluation of Selected Proteins and Quantification by Digital Image Analysis

The selected differentially expressed proteins that were identified by the LFQ nLC-MS/MS-based proteomic analysis were immunohistochemically evaluated on whole biopsy FFPE tumor tissue sections; this was comprised from the MPN-AITL/R-AITL comparison: isocitrate dehydrogenase 2 (*IDH2*), DnaJ homolog subfamily A member 2 (*DNAJA2*), and citrate synthase; and from the MPN-DLBCL/R-DLBCL comparison: lactotransferrin and myeloblastin. A detailed description is given in the Supplementary methods. Immunohistochemical staining was quantified by digital image analysis, and area fractions (AFs), which are defined as the stained area of each tissue section normalized to the area of interest, were compared between MPN-lymphoma and reference samples [18,25,26,27]. Expression levels of *IDH2* protein were based on AFs of strong and intermediate intensity staining, and expression levels of *DNAJA2* protein, citrate synthase, lactotransferrin, and myeloblastin were based on AFs of all positive staining.

### 2.4. Statistical Analysis

A student’s t-test was used for the statistical analysis of the fold changes of differentially expressed proteins. Pathway analysis was performed with the identified proteins entered with gene names, log_2_ fold-changes and *p*-values into Ingenuity Pathway Analysis (Canonical Pathway, QIAGEN Inc., Hilden, Germany, https://www.qiagenbioinformatics.com/products/ingenuitypathway-analysis, 7 June 2020) [24]. When entered, 1074 and 1141 genes were recognised from the AITL and the DLBCL analysis, respectively.

The differences in clinicopathological features were assessed using Fisher’s exact test. With proteomic analyses, missing values will most often be present. For the principal component analysis (PCA), in the few cases of missing values, the median expression of the protein from the remaining samples was used. Using the lowest, median, and largest values, respectively, did not notably change the placement of samples in the PCA plots. Differences in AFs between MPN-lymphoma and reference samples were assessed using an independent Mann–Whitney U test. *p*-values < 5% were considered statistically significant. Statistical analyses were performed using R Statistical Software (version 4.0.2, RStudio: Intergrated Development Environment for R, Boston, MA, USA, http://www.rstudio.com, 7 June 2020).

## 3. Results

### 3.1. Description of the Study Cohort

The patient cohort consisted of 34 patients, including age and tissue matched controls (Table 1). The MPN diagnoses included ET, PV, PMF, and MPN-U. Table 1 summarizes the clinical features of the patients; the groups were comparable according to sex, age, and international prognostic index.

### 3.2. Differentially Expressed Proteins Identified between Lymphoma Specimens from Patients with/without MPN

The protein expression in the diagnostic lymphoma samples from all of the included patients was assessed by proteomics. In total, a combined set of 3083 proteins were identified in the 34 samples that were analysed. In the analysis of lymphoma biopsies from MPN-AITL vs. R-AITL patients, a total of 1074 proteins were present in at least 70% of the samples in each group and 20 proteins were identified as significantly differentially expressed (Table 2, Figure 1A). Of those 20, seven proteins were upregulated in the MPN-AITL specimens, including *DNAJA2* (fold change 1.5), and 13 proteins were downregulated in the MPN-AITL samples, including *IDH2* and citrate synthase (fold changes both 0.6). Immunohistochemical evaluation of *IDH2*, *DNAJA2* and citrate synthase showed diffuse cytoplasmatic staining of both neoplastic and non-neoplastic cells in the tumor microenvironment in both MPN-AITL and R-AITL samples (Figure 2A,D,G). Quantification of immunohistochemical staining of both tumor cells and the tumor microenvironment showed a slight tendency towards higher levels (as opposed to lower levels in the proteomic analysis) of *IDH2* protein expression in MPN-AITL samples compared with R-AITL samples (*p* = 0.105) (Figure 2B). In particular, lymphoma samples with concurrent myeloid disease showed high variations in *IDH2* expression compared with the R-AITL samples, which was consistent with the high differences in expression that were also found in the proteomic analysis (Figure 2C). For both *DNAJA2* and citrate synthase, an immunohistochemical evaluation showed differential protein expression, with both being significantly increased in the MPN-AITL compared with the R-AITL tumors (*p* = 0.015 and *p* = 0.007, respectively) (Figure 2E,H).

In the B-cell lymphomas, a total of 1141 proteins were present in at least 70% of the samples in each group; 34 proteins were identified as significantly differentially expressed between the lymphomas from MPN-DLBCL and R-DLBCL patients (Table 3, Figure 1B). Of these, 16 proteins were upregulated in the MPN-DLBCL specimens, including myeloblastin (also known as proteinase 3, *PRTN3*) and lactotransferrin, the latter being markedly upregulated (fold changes 4.4 and 24.4, respectively). Moreover, 18 proteins were downregulated in the MPN-DLBCL specimens, including nine different ribosomal proteins (fold change range 0.6–0.8). As in the proteomics analysis, an immunohistochemical evaluation revealed a high variation in lactotransferrin expression in the MPN-DLBCL group. Nonetheless, based on its localization in the cytoplasm in restricted cellular subsets, lactotransferrin retained its higher expression in MPN-DLBCL compared with R-DLBCL (*p* = 0.050) (Figure 2J,K). This difference was mainly due to the high expression levels in the three patient samples, which correlate with the high variation in expression levels that was found in the proteomics analysis (Figure 2L). In contrast, after evaluating myeloblastin with immunohistochemistry, no significant differential expression was observed between MPN-DLBCL and R-DLBCL sections (*p* = 0.297) (Figure 2M,N). The expression pattern of myeloblastin revealed a slightly diffuse cytoplasmatic staining that was restricted to certain cell types within the tumor, with a wide distribution of expression levels that correlate to the differences in expression levels that were found in the proteomics analysis (Figure 2O).

In both the MPN-AITL/R-AITL and MPN-DLBCL/R-DLBCL analyses, almost all of the identified proteins presented with overlapping expression levels between the MPN-lymphomas and references. Moreover, in the immunohistochemical evaluation in particular, several proteins presented with a relatively wide distribution of expression levels, especially when a myeloid component was present in combination with a lymphoma (Figure 2 and Appendix A). In contrast, in lymphoma samples without MPN, the expression levels of most proteins were more consistent among the patient samples. An example was lactotransferrin, which had a high fold change (24-fold), although among the examined samples, four MPN-DLBCL exhibited very high expression, while the other five MPN-DLBCL showed expression levels similar to the R-DLBCL (Figure 2L and Appendix A).

### 3.3. Differentially Expressed Proteins Distinguish Lymphomas with and without MPN

Unsupervised PCA with the input of the 20 significantly differentially expressed proteins showed distinct clustering that corresponded to the MPN-AITL and R-AITL, respectively (Figure 1C and Appendix A). The pathway analysis revealed disturbances in several pathways (Table 4A, Figure 3). These pathways include the citric acid cycle, amino acid metabolism, and the responses to cellular stress. This suggests that biological differences may be seen in pathogenesis when myeloproliferative and lymphoproliferative neoplasia exist concurrently, thereby influencing cellular respiration, energy metabolism, biosynthetic processes, and cellular stress.

Similarly, analysing the MPN-DLBCL and R-DLBCL specimens by PCA with the input of the 34 significantly differentially expressed proteins revealed an almost distinct clustering, with two of the samples, one MPN-DLBCL and one R-DLBCL, showing closer clustering to the opposite group (Figure 1D and Appendix A). The pathway analysis revealed disturbances in the G protein and GTPase signalling, mammalian target of rapamycin (mTOR) signalling, phospholipase C (*PLC*) signalling, and a significant downregulation of translation initiation (Table 4B, Figure 4). This implies that pathogenetic differences with regard to a concurrent MPN may be present, influencing translation of mRNA to polypeptides and G protein-coupled signal transduction. Furthermore, the ingenuity pathway analysis indicated by inference that the dysregulation of three upstream regulators, *MLXIPL*, *MYCN*, and *RICTOR*, may explain the differences that were observed in the expression of several of the ribosomal proteins between MPN-DLBCL and R-DLBCL specimens (Table 4C, Appendix A).

### 3.4. Shared Proteomic Differences between MPN-AITL and MPN-DLBCL

Based on the hypothesis that MPN and lymphoma may share genomic alterations, which represents a possible pathogenetic relationship, we sought to investigate whether the protein expression pattern in lymphomas from MPN-AITL and MPN-DLBCL specimens showed any resemblances that linked MPN to lymphoma that were independent of the B- or T-cell origin.

Interestingly, none of the significantly differentially expressed proteins that were identified between T-cell and B-cell lymphomas from patients with and without a concurrent MPN were shared. However, the proteomic analysis showed disturbances in eukaryotic translation initiation in both the MPN-AITL and MPN-DLBCL samples. Two proteins of eukaryotic translation initiation (*EIF2S2* and *EIF6*) were identified as significantly increased in the samples from MPN-AITL compared with AITL patients. These two proteins play a major role in one of the significantly disturbed pathways, i.e., EIF2 signalling and thus eukaryotic translation initiation, which is one of the significantly downregulated pathways that was identified between the MPN-DLBCL and R-DLBCL tumors.

## 4. Discussion

We present the first study of proteomic-based analyses showing differences in protein expression between tumor tissues from lymphoma patients with or without concurrent MPN. Our findings support the hypothesis that there may be an association between the development of MPN and lymphoma in these patients.

Currently, the lymphoma component from patients with and without MPN cannot be distinguished by histomorphology and no known biomarkers are able to differentiate the MPN lymphomas and reference lymphomas. In the present study, we searched for molecular differences at the protein level in the lymphoma tissue, and the MS-based proteomics analysis identified subtle but significantly differentially expressed proteins between the MPN-AITL and R-AITL tumor tissue, as well as between the MPN-DLBCL and R-DLBCL specimens. Interestingly, based on the 20 significantly differentially expressed proteins between MPN-AITL and R-AITL specimens, the PCA analysis identified two distinct patterns in protein composition that corresponded to the MPN-AITL and R-AITL tumors, respectively. In the case of MPN-DLBCL/R-DLBCL, the proteomic profiles enabled the almost distinct clustering by PCA between MPN-DLBCL and R-DLBCL. While the differences that were found in the protein expression reflect a rather subtle biological diversity, they enabled the clustering into distinct groups. Thus, our results support the hypothesis that molecular differences are present between lymphomas from patients with and without MPN. The differences that were observed between MPN- and non-MPN-associated lymphomas are subtle, yet important, as reflected by the inferior survival that was reported previously. Although promising treatment strategies exist for both AITL and DLBCL patients, the therapeutic interventions may be compromised by the differences in the underlying tumor biology in the MPN-associated lymphomas, ultimately leading to poorer prognosis.

In addition, although hypothetical, our findings are compatible with the hypothesis that the lymphomas develop through different pathological mechanisms, since the proteomes in the two lymphomas are different, thereby representing two distinct biological diseases. Whereas AITL and DLBCL originates from mature lymphocytes [9,12], it could be postulated that the development of MPN-lymphomas may descend from earlier hematopoietic progenitors harboring genetic alterations that later drive the development of both malignancies. In that case, pathogenetic mechanisms may relate back to early events occurring in the hematopoietic stem cells, and the clonal expansion of the cells descending from an abnormal hematopoietic progenitor cell may be responsible for the development of both MPN and lymphoma. Further data supporting this hypothesis are warranted.

Notably, *IDH2* and citrate synthase were found to be differentially expressed in MPN-AITL compared with R-AITL biopsies in the proteomics analysis. Both are enzymes that are involved in the citric acid cycle, cellular respiration and energy metabolism [28,29,30]. Additionally, amino acid substitutions based on *IDH2* gene mutations, leading to mutant *IDH2* enzymes, causes the conversion of the normal product α-ketoglutarate to 2-hydroxyglutarate (2HG), which has been shown to act as an oncometabolite and which has been suggested to be responsible for driving tumor progression [31,32]. *IDH2* protein mutants causing elevated 2HG are known to occur in several malignancies, including acute myeloid leukemia and AITL [28,31].

Mutations in the *IDH2* gene are well-known in AITL [33,34]. *IDH2* gene mutation status is available from the lymphomas of five of the included MPN-AITL patients [6]. One of the MPN-AITL samples in the present study harbored two different *IDH2* gene mutations (R172G and R140Q) and was found with barely perceptible *IDH2* protein expression, Figure 2B. As the anti-*IDH2* antibody used in the present study recognizes wildtype (wt) *IDH2* protein, it could be postulated that in this sample, almost no wt *IDH2* protein was expressed. If this patient, together with their matched R-AITL reference, were excluded, then *IDH2* was significantly differentially expressed between the two patient groups (*p* = 0.026). Interestingly, the two other samples with a known *IDH2* mutation (R172M) were found within the three samples that had the highest expression level of wt *IDH2* [6]. Unfortunately, an evaluation of the *IDH2* gene mutation status of the entire MPN-lymphoma cohort was not possible because of unavailability (or poor quality) of the tissue specimens that were acquired for sequencing. Thus, when evaluating *IDH2* protein expression and its possible impact on cellular mechanisms, both mutant and wt *IDH2* protein must be taken into consideration in order to gain insight into the presumed pathogenic association of *IDH2* in myeloid and lymphoid disorders in general, and in MPN-AITL patients in particular.

Additionally, the proteomic analysis showed an increased protein expression of *DNAJA2* in MPN-AITL tumor tissue compared with R-AITL. The function of *DNAJA2* is not fully elucidated, however heat shock proteins, including different *DNAJA*s, have been found to be overexpressed in cancers, resulting in antiapoptotic activity contributing to cancer cell survival [35], which could ultimately aid the development of both MPN and AITL in the patients.

In the MPN-DLBCL compared with the R-DLBCL samples, both myeloblastin and lactotransferrin were identified as upregulated. Myeloblastin is a myeloid specific serine protease that has been found upregulated in hematopoietic progenitor cells during myeloid differentiation. It has been found to be overexpressed in myeloid leukemia and clinical studies have shown worse clinical outcomes when the protease is present in the tumor microenvironment [36,37,38,39]. Lactotransferrin is an iron-binding glycoprotein that is involved in iron homeostasis and immunomodulation. Furthermore, it has been shown to reduce tumor growth through the regulation of cellular growth and differentiation, and to induce apoptosis in cancer cells [40,41].

Pathway analysis identified the various pathways that are possibly disturbed as a result of the identified altered protein expression. In MPN-AITL/R-AITL specimens, several of the pathways indicate differences in cellular respiration and oxidative metabolism between the two patient groups. The citric acid cycle, in conjunction with oxidative phosphorylation, provides the vast majority of the energy used by aerobic cells, and is the most important central pathway, linking together almost all metabolic pathways [30,42]. Notably, unlike other metabolic pathways, only very few genetic abnormalities of the citric acid cycle are known, probably because of the vital importance of the cycle, showing the impact of altered protein expression influencing the cycle [30].

Regarding MPN-DLBCL/R-DLBCL analysis, several of the identified pathways affect mechanisms of translation initiation and G protein/GTPase, mTOR, and PLC signalling. Translation initiation is highly dependent on both eukaryotic initiation factors (eIFs) and guanosine triphosphate (GTP) hydrolysis [43]. In general, G protein-coupled receptor (GPCR) signalling is involved in a wide variety of cellular processes, including intracellular signal transduction, resulting in, e.g., the production of secondary messenger molecules and activation of the MAPK and PLC pathways. In addition, it has been found to be involved in the growth and metastasis of different tumors [44,45].

We found discrepancies in protein expression levels of *IDH2*, citrate synthase, and myeloblastin when analyzed by mass spectrometry quantifying on peptide level and when analyzed on cellular level with immunohistochemistry. These differences may be explained by differences in methodologies. The MS-based proteomics approach is a large-scale method that analyses a large number of peptides, from which it identifies protein families based on peptide sequences. Interestingly, for all differentially expressed proteins, a high variation in expression levels was observed between specimens. The proteomic analyses were performed on whole tissue sections, thus neoplastic as well as non-neoplastic cells of the tumor microenvironment from both lymphoid and non-lymphoid tissues were taken into account. On the other hand, the staining quantification in the immunohistochemical analyses was confined to the ROI, thereby evaluating the neoplastic cells and the surrounding tumor microenvironment, restricted to lymphoid tissue of the biopsies.

In this study, we show that the protein expression of *IDH2* in MPN-AITL is different from that found in R-AITL, but more functional studies are warranted in order to decide the impact of the *IDH2* protein changes, ideally in combination with an analysis of the *IDH2* gene mutational status. Additionally, we found a higher variation in protein expression in the samples with a concurrent myeloid component compared to those without. Although the biological mechanisms behind the pathology of MPN-lymphomas are far from being fully elucidated, this hypothesis-generating study reveals the differences in protein expressions that give clues to the origin of lymphomas with concurrent MPN. This patient cohort is truly heterogeneous, which is reflected both in the differences in cell composition in the lymphoma tumor tissue as well as in the variations in the MPN component, e.g., blood count and putative hypoxia caused by significant anemia or MPN therapy at the time of the lymphoma diagnosis (at sampling). This heterogeneous setting may affect the protein profile and as described, these results must be taken as hypothesis-generating and await further validation in additional larger and independent cohorts. This study was also based on a small sample cohort, which poses a challenge in protein profiling and subsequent statistical data analysis. However, as the combination of MPN and lymphoma is infrequent, this cohort is the largest published of its kind, and no other studies have shown differences in the proteome from comparing MPN-lymphomas and reference lymphomas.

## 5. Conclusions

Lymphoma patients with or without MPN show subtle but important differences in the protein expression that enables clustering of the lymphomas, thus indicating differences between the malignancies at the molecular level. This is the first study to provide MS-based protein profiling in patients with both MPN and lymphoma.

## Figures and Tables

**Figure 1 cancers-13-05526-f001:**
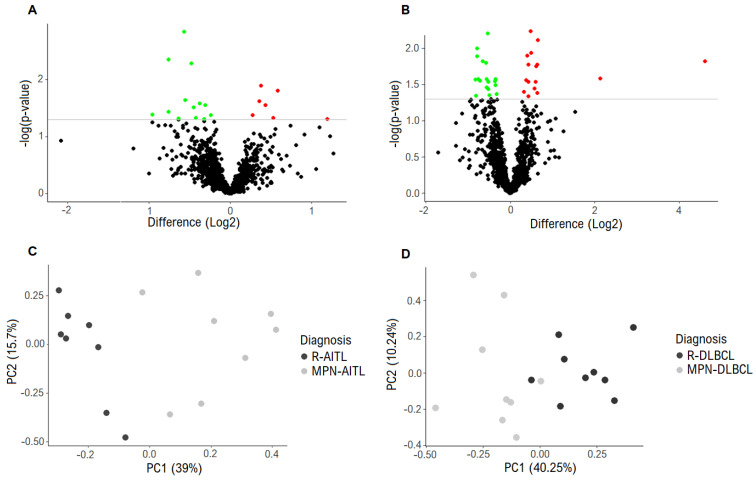
Differentially expressed proteins. (**A**) Proteomic analysis revealed 20 significantly differentially expressed proteins between the MPN-AITL and R-AITL biopsies. The grey horizontal line marks the threshold of *p* < 0.05 of significantly expressed proteins. Green: downregulated. Red: upregulated. (**B**) Proteomic analysis showed 34 significantly differentially expressed proteins between the MPN-DLBCL and R-DLBCL biopsies. The grey horizontal line marks the threshold of *p* < 0.05 of significantly expressed proteins. Green: downregulated. Red: upregulated. (**C**) PCA with input of the 20 differentially expressed proteins revealed distinct clustering between the MPN-AITL and R-AITL specimens. Dark grey represents the R-AITL samples and light grey represents the MPN-AITL samples. (**D**) PCA with input of the 34 differentially expressed proteins showed clustering between the MPN-DLBCL and R-DLBCL specimens, with the exception of two samples. Dark grey represents R-DLBCL samples and light grey represent MPN-DLBCL samples. Abbreviations: AITL, angioimmunoblastic T-cell lymphoma; DLBCL, diffuse large B-cell lymphoma; MPN, myeloproliferative neoplasia; PCA, principal component analysis; R-, reference sample.

**Figure 2 cancers-13-05526-f002:**
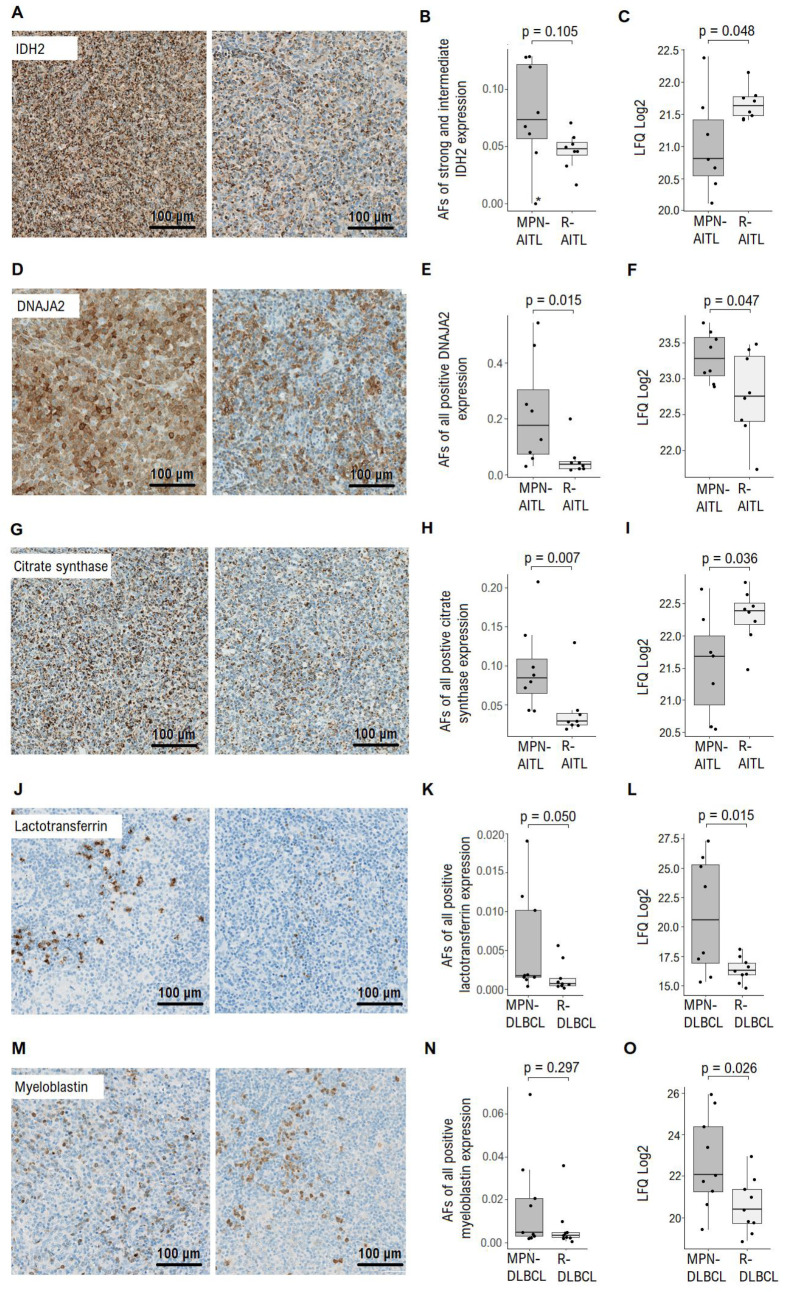
Immunohistochemical evaluation of selected proteins. (**A**) Representative images of *IDH2* protein staining expression (**B**) AFs of strong and intermediate intensity staining of *IDH2* protein. * denote MPN-AITL patient with two known *IDH2* gene mutations. (**C**) *IDH2* protein expression identified by MS-based proteomics analysis. (**D**) Representative images of *DNAJA2* staining expression. (**E**) AFs of all positive *DNAJA2* protein staining. (**F**) *DNAJA2* protein expression identified by MS-based proteomics analysis. (**G**) Representative images of citrate synthase staining expression. (**H**) AFs of all positive citrate synthase staining. (**I**) Citrate synthase protein expression identified by MS-based proteomics analysis. (**J**) Representative images of lactotransferrin staining expression. (**K**) AFs of all positive lactotransferrin staining. (**L**) Lactotransferrin protein expression identified by MS-based proteomics analysis. (**M**) Representative images of myeloblastin staining expression. (**N**) AFs of all positive myeloblastin staining. (**O**) Myeloblastin protein expression identified by MS-based proteomics analysis. Abbreviations: AF, area fraction; AITL, angioimmunoblastic T-cell lymphoma; DLBCL, diffuse large B-cell lymphoma; *DNAJA2*, DnaJ homolog subfamily A member 2; *IDH2*, isocitrate dehydrogenase 2; LFQ, label free quantification; MPN, myeloproliferative neoplasia; R-, reference sample.

**Figure 3 cancers-13-05526-f003:**
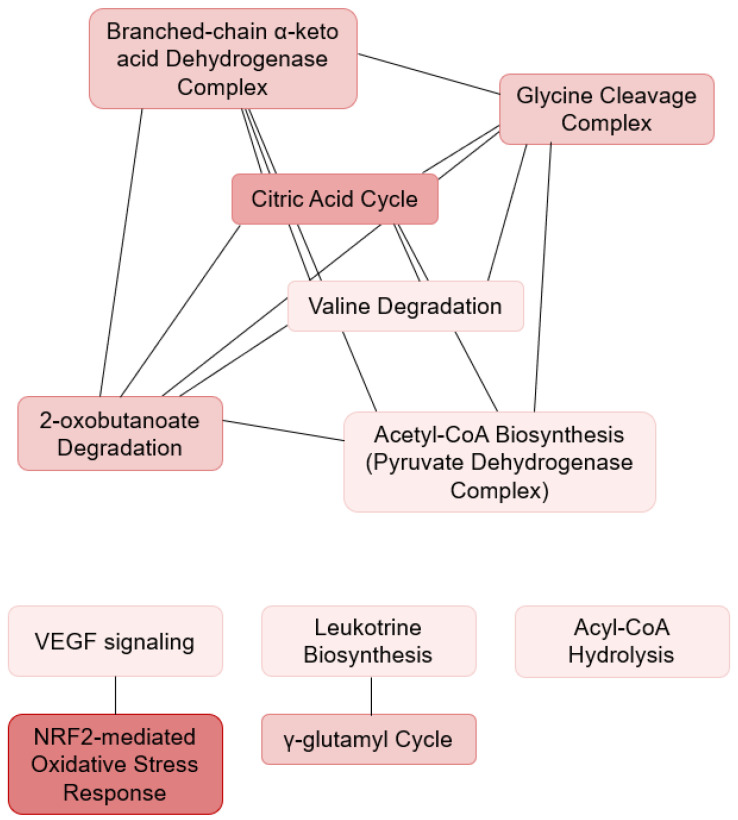
Ingenuity pathway analysis in MPN-AITL and R-AITL patients. Significantly disturbed pathways between the MPN-AITL and R-AITL specimens showing shared proteins between the pathways. Abbreviations: *VEGF*, Vascular Endothelial Growth Factor; *NFR2*, nuclear factor erythroid 2-like 2.

**Figure 4 cancers-13-05526-f004:**
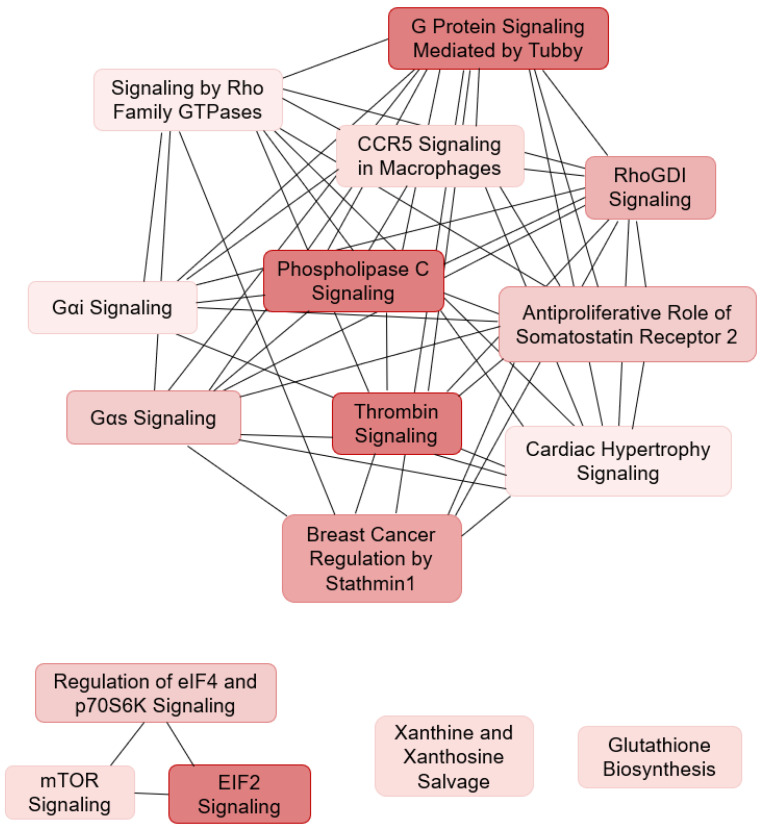
Ingenuity pathway analysis in MPN-DLBCL and R-DLBCL patients. Significantly disturbed pathways between the MPN-DLBCL and R-DLBCL specimens showing shared proteins between the pathways. Abbreviations: GTP, guanosine triphosphate; *CCR5*, C-C chemokine receptor type 5; RhoGDI, Rho GDP-dissociation inhibitor; eIF4, eukaryotic initiation factor 4; p70S6K, Ribosomal protein S6 kinase beta-1; eIF2, eukaryotic initiation factor 2; mTOR, The mammalian target of rapamycin.

**Table 1 cancers-13-05526-t001:** Clinicopathological features.

	MPN-AITL	R-AITL		MPN-DLBCL	R-DLBCL	
Clinicopathological Feature	*n* = 8	*n* = 8	*p*-Value	*n* = 9	*n* = 9	*p*-Value
**Age at diagnosis, y**			NS			NS
Median	71	67		66	68	
Range	62–89	57–87		58–84	47–81	
**Sex**			NS			NS
Male	4	3		6	7	
Female	4	5		3	2	
**IPI**			NS			NS
Low	0	1		3	3	
Low-intermediate	2	3		3	3	
High-intermediate	3	3		2	2	
High	3	1		1	1	
***IDH2* protein expression**			0.105			0.796
Above median	6	2		5	4	
Below median	2	6		4	5	
***DNAJA2* protein expression**			**0.015**			0.730
Above median	6	2		4	5	
Below median	2	6		5	4	
**Citrate synthase expression**			**0.007**			0.387
Above median	7	1		6	3	
Below median	1	7		3	6	
**Lactotransferrin expression**			0.645			**0.050**
Above median	5	3		7	2	
Below median	3	5		2	7	
**Myeloblastin expression**			0.105			0.297
Above median	4	4		5	4	
Below median	4	4		4	5	

Cutoffs for protein expression levels were based on the median expression of the respective protein in all samples. *p*-values on protein expression are based on the immunohistochemical evaluation. Significant (*p* < 0.05) and trending differences are marked in bold. Abbreviations: AITL, angioimmunoblastic T cell lymphoma; DLBCL, diffuse large B-cell lymphoma; *DNAJA2*, DnaJ homolog subfamily A member 2; IDH2, isocitrate dehydrogenase 2; IPI, International Prognostic Index; MPN, myeloproliferative neoplasia; NS, not significant; R-, reference sample.

**Table 2 cancers-13-05526-t002:** Differentially expressed proteins between MPN-AITL and R-AITL patients.

	Fold Change (MPN-AITL/ R-AITL)	Protein Name	Gene Name	*p*-Value
**Upregulated**	2.3	Ig gamma-4 chain C region	*IGHG4*	0.049
	1.5	Signal recognition particle 14 kDa protein	*SRP14*	0.016
	1.5	DnaJ homolog subfamily A member 2 *	*DNAJA2*	0.047
	1.4	Oxysterol-binding protein 1	*OSBP*	0.028
	1.3	Eukaryotic translation initiation factor 2 subunit 2	*EIF2S2*	0.013
	1.3	UMP-CMP kinase	*CMPK1*	0.024
	1.2	Eukaryotic translation initiation factor 6	*EIF6*	0.042
**Downregulated**	0.5	Actin, aortic smooth muscle, Actin, gamma-enteric smooth muscle	*ACTA2*,*ACTG2*	0.041
	0.6	Cytoplasmic FMR1-interacting protein 1	*CYFIP1*	0.004
	0.6	Citrate synthase, mitochondrial *	*CS*	0.036
	0.6	Isocitrate dehydrogenase [NADP], mitochondrial *	*IDH2*	0.048
	0.7	Glutathione hydrolase 5 proenzyme	*GGT5*	0.001
	0.7	V-type proton ATPase subunit E 1	*ATP6V1E1*	0.005
	0.7	Dihydrolipoyl dehydrogenase, mitochondrial	*DLD*	0.023
	0.7	Palmitoyl-protein thioesterase 1	*PPT1*	0.031
	0.7	Glutathione S-transferase P	*GSTP1*	0.047
	0.8	Aspartyl aminopeptidase	*DNPEP*	0.026
	0.8	Dynactin subunit 1	*DCTN1*	0.028
	0.8	V-type proton ATPase catalytic subunit A	*ATP6V1A*	0.042
	0.8	DnaJ homolog subfamily C member 13	*DNAJC13*	0.049

A total of 20 proteins were identified as significantly differentially expressed between diagnostic lymphoma samples from MPN-AITL and R-AITL patients. * Protein expression evaluated by immunohistochemistry. Abbreviations: AITL, angioimmunoblastic T-cell lymphoma; MPN, myeloproliferative neoplasia; R-, reference sample.

**Table 3 cancers-13-05526-t003:** Differentially expressed proteins between MPN-DLBCL and R-DLBCL patients.

	Fold Change (MPN-DLBCL /R-DLBCL)	Protein Name	Gene Name	*p*-Value
**Upregulated**	24.4	Lactotransferrin *	*LTF*	0.015
	4.4	Myeloblastin *	*PRTN3*	0.026
	1.6	Endophilin-B1	*SH3GLB1*	0.008
	1.6	Alpha-1-antitrypsin	*SERPINA1*	0.017
	1.6	ADP/ATP translocase 3	*SLC25A6*	0.041
	1.5	Mannose-1-phosphate guanyl transferase beta	*GMPPB*	0.017
	1.5	Echinoderm microtubule-associated protein-like 2	*EML2*	0.028
	1.5	Band 4.1-like protein 2	*EPB41L2*	0.036
	1.4	Guanine nucleotide-binding protein G(I)/G(S)/G(T) subunit beta-2	*GNB2*	0.006
	1.4	Deoxyribose-phosphate aldolase	*DERA*	0.011
	1.3	COMM domain-containing protein 10	*COMMD10*	0.013
	1.3	Neuroblast differentiation-associated protein AHNAK	*AHNAK*	0.017
	1.3	Glutathione synthetase	*GSS*	0.027
	1.3	Calpastatin	*CAST*	0.029
	1.3	Guanine nucleotide-binding protein G(I)/G(S)/G(T) subunit beta-1	*GNB1*	0.039
	1.3	Annexin A2, Putative annexin A2-like protein	*ANXA2*, *ANXA2P2*	0.045
**Downregulated**	0.6	Ribose-5-phosphate isomerase	*RPIA*	0.010
	0.6	Replication protein A 14 kDa subunit	*RPA3*	0.013
	0.6	40S ribosomal protein S13	*RPS13*	0.015
	0.6	Rho guanine nucleotide exchange factor 1	*ARHGEF1*	0.027
	0.6	40S ribosomal protein S19	*RPS19*	0.027
	0.6	60S ribosomal protein L38	*RPL38*	0.028
	0.6	Glutaredoxin-3	*GLRX3*	0.045
	0.7	WD repeat-containing protein 61	*WDR61*	0.006
	0.7	60S ribosomal protein L27a	*RPL27A*	0.016
	0.7	60S ribosomal protein L24	*RPL24*	0.026
	0.7	40S ribosomal protein S24	*RPS24*	0.029
	0.7	Rho guanine nucleotide exchange factor 2	*ARHGEF2*	0.034
	0.7	Purine nucleoside phosphorylase	*PNP*	0.036
	0.8	60S ribosomal protein L9	*RPL9*	0.026
	0.8	40S ribosomal protein S12	*RPS12*	0.028
	0.8	40S ribosomal protein SA	*RPSA*	0.032
	0.8	Adenylosuccinate synthetase isozyme 2	*ADSS*	0.042
	0.8	Zinc finger CCCH-type antiviral protein 1	*ZC3HAV1*	0.044

A total of 34 proteins were identified as significantly differentially expressed between diagnostic lymphoma samples from MPN-DLBCL and R-DLBCL patients. * Protein expression evaluated by immunohistochemistry. Abbreviations: DLBCL, diffuse large B cell lymphoma; MPN, myeloproliferative neoplasia; R-, reference sample.

**Table 4 cancers-13-05526-t004:** Disturbed pathways.

(A)	Pathway Name			Molecules	*p*-Value
	NRF2-mediated Oxidative Stress response			*ACTA2*, *DNAJA2*, *DNAJA13*, *GSTP1*	0.003
	TCA cycle II (Eukaryotic)			*CS*, *DLD*	0.011
	Branched-chain α-keto acid Dehydrogenase Complex			*DLD*	0.019
	Glycine Cleavage Complex			*DLD*	0.019
	γ-glutamyl Cycle			*GGT5*	0.019
	2-oxobutanoate Degradation I			*DLD*	0.019
	Acyl-CoA Hydrolysis			*PPT1*	0.037
	Leukotriene Biosynthesis			*GGT5*	0.037
	Acetyl-CoA Biosynthesis I (Pyruvate Dehydrogenase Complex)			*DLD*	0.037
	Valine Degradation I			*DLD*	0.037
	VEGF Signalling			*ACTA*, *EIF2S2*	0.038
**(B)**	**Pathway Name**			**Molecules**	***p*-Value**
	* EIF2 Signalling			*RPL24*, *RPL27A*, *RPL38*, *RPL9*, *RPS12*, *RPS13*, *RPS19*, *RPS24*, *RPSA*	0.001
	Phospholipase C Signalling			*AHNAK*, *ARHGEF1*, *ARHGEF2*, *GNB1*, *GNB2*	0.004
	Thrombin Signalling			*ARHGEF1*, *ARHGEF2*, *GNB1*, *GNB2*	0.005
	G Protein Signalling Mediated by Tubby			*GNB1*, *GNB2*	0.005
	Breast Cancer Regulation by Stathmin1			*ARHGEF1*, *ARHGEF2*, *GNB1*, *GNB2*	0.013
	RhoGDI Signalling			*ARHGEF1*, *ARHGEF2*, *GNB1*, *GNB2*	0.017
	Antiproliferative Role of Somatostatin Receptor 2			*GNB1*, *GNB2*	0.022
	Gαs Signalling			*GNB1*, *GNB2*	0.022
	Regulation of eIF4 and p70S6K Signalling			*RPS12*, *RPS13*, *RPS19*, *RPS24*, *RPSA*	0.025
	CCR5 Signalling in Macrophages			*GNB1*, *GNB2*	0.028
	Glutathione Biosynthesis			*GSS*	0.030
	Xanthine and Xanthosine Salvage			*PNP*	0.030
	mTOR Signalling			*RPS12*, *RPS13*, *RPS19*, *RPS24*, *RPSA*	0.032
	Cardiac Hypertrophy Signalling			*ADSS2*, *GNB1*, *GNB2*	0.036
	Gαi Signalling			*GNB1*, *GNB2*	0.041
	Signalling by Rho Family GTPases			*ARHGEF1*, *ARHGEF2*, *GNB1*, *GNB2*	0.042
**(C)**	**Upstream Regulator**	**Predicted Activation State**	***p*-Value**	**Target Molecules**	
	MLXIPL	Inhibited	<0.001	*RPL24*, *RPL27A*, *RPL38*, *RPL9*, *RPS12*, *RPS13*, *RPS19*, *RPS24*, *RPSA*	
	MYCN	Inhibited	0.009	*RPL24*, *RPL27A*, *RPL38*, *RPL9*, *RPS12*, *RPS13*, *RPS19*, *RPS24*	
	RICTOR	Activated	0.042	*RPL38*, *RPL9*, *RPS13*, *RPS19*, *RPS24*, *RPSA*	

Ingenuity Pathway Analysis. (**A**) Significantly disturbed pathways between MPN-AITL and R-AITL tumors based on all the differentially expressed proteins identified by proteomics. (**B**) Significantly disturbed pathways between MPN-DLBCL and R-DLBCL tumors based on all the differentially expressed proteins identified by proteomics. * The pathway is significantly inhibited, z-score: -2. (**C**) Pathway analysis revealed three upstream regulators whose alteration may be explanatory for differences observed in specific target molecules between tumors from MPN-DLBCL and R-DLBCL patients. Abbreviations: AITL, angioimmunoblastic T cell lymphoma; DLBCL, diffuse large B-cell lymphoma; MPN, myeloproliferative neoplasia; R-, reference sample.

## Data Availability

These data analyzed during the current study are available upon reasonable request.

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
