# Peer review of "Proteomic Profiling Differentiates Lymphoma Patients with and without Concurrent Myeloproliferative Neoplasia"

_cancers, 2021, doi:10.3390/cancers13215526_

Round 1

Reviewer 1 Report

In the submitted manuscript "Proteomic profiling differentiates lymphoma patients with and  without concurrent myeloproliferative neoplasia" the authors describe the  differences in protein expression that enable clustering
of the MPN and non-MPN "associated" lymphomas. It seems that the differences are subtle, nevertheless important.  In my opinion this opens a new "window" in the diagnosis, treatment and prognosis of these patients. The manuscript is carefully prepared, the results are clear, the discussion is comprehensive. From a biochemical point of view I would like to ask the authors to express with a few sentences  their hypothesis why such a difference exists and  why those patients (in authors' opinion) have an inferior survival despite  optimal treatment.  I do not have anything to add to this presentation. I do recommend accepting the manuscript for publication in the present form after minor suggested changes.

Author Response

Point-by point response to reviewer comments, Cancers-1429931_Reviewer 1

Reviewer 1:

In the submitted manuscript "Proteomic profiling differentiates lymphoma patients with and without concurrent myeloproliferative neoplasia" the authors describe the differences in protein expression that enable clustering of the MPN and non-MPN "associated" lymphomas. It seems that the differences are subtle, nevertheless important.  In my opinion this opens a new "window" in the diagnosis, treatment and prognosis of these patients. The manuscript is carefully prepared, the results are clear, the discussion is comprehensive. From a biochemical point of view, I would like to ask the authors to express with a few sentences their hypothesis why such a difference exists and why those patients (in authors' opinion) have an inferior survival despite optimal treatment.  I do not have anything to add to this presentation. I do recommend accepting the manuscript for publication in the present form after minor suggested changes.

Response: We thank the reviewer for the suggested discussion point which we fully agree with is an interesting view on these patients and may reflect altered underlying biology.

In the discussion section, we have already commented on our hypothesis leading to the difference between MPN and non-MPN associated lymphomas: “In addition, although hypothetical, our findings are compatible with the hypothesis that the lymphomas develop through different pathological mechanisms in that the proteome in the two lymphomas are different, thereby representing two distinct biological diseases. Whereas AITL and DLBCL originates from mature lymphocytes [9,12], it could be postulated that the development of MPN-lymphomas may descend from earlier hematopoietic progenitors harboring genetic alterations that later drive the development of both malignancies. In that case, pathogenetic mechanisms may relate back to early events occurring in the hematopoietic stem cells, and the clonal expansion of cells descending from an abnormal hematopoietic progenitor cell may be responsible for the development of both MPN and lymphoma. Further data supporting this hypothesis are warranted.”

We have added the following in the discussion section: “The differences observed between the MPN and non-MPN associated lymphomas are subtle, however important as reflected by the inferior survival reported previously [6]. Although promising treatment strategies exist for both AITL and DLCBL patients, the therapeutic interventions may be compromised by differences in the underlying tumor biology in the MPN associated lymphomas, ultimately leading to poorer prognosis.”

Reviewer 2 Report

The goal of the manuscript is to describe and interpret lymphoma tissue proteomic profile of patients with or without concurrent myeloproliferative neoplasia. To my opinion the sample processing, proteomic analysis and bioinformatics analysis is well executed. From the clinical point of view the presented myeloproliferative diseases (ET, PV, PMF, and MPN-U) may present with significant variations in blood count among each other and also this kind of variability may not be seen in patients with lymphomas, e.g. severe anemia vs. significant polycythemia.  Also some patients with MPN diagnosed before the diagnosis of lymphoma might have been already on therapy. Since the hypoxia caused by significant anemia or the therapy at the time of sample collection may affect the protein profile authors should address these in results or discussion.

Author Response

Point-by point response to reviewer comments, Cancers-1429931

Reviewer 2:

The goal of the manuscript is to describe and interpret lymphoma tissue proteomic profile of patients with or without concurrent myeloproliferative neoplasia. To my opinion the sample processing, proteomic analysis and bioinformatics analysis is well executed. From the clinical point of view the presented myeloproliferative diseases (ET, PV, PMF, and MPN-U) may present with significant variations in blood count among each other and also this kind of variability may not be seen in patients with lymphomas, e.g severe anemia vs. significant polycythemia.  Also, some patients with MPN diagnosed before the diagnosis of lymphoma might have been already on therapy. Since the hypoxia caused by significant anemia or the therapy at the time of sample collection may affect the protein profile authors should address these in results or discussion.

Response: We agree with the reviewer and have added the discussion point in the discussion section: “This patient cohort is truly heterogeneous reflected both in differences in cell composition in the lymphoma tumor tissue as well as variations in the MNP component, e.g., blood count as well as putative hypoxia caused by significant anemia or MPN therapy at time of lymphoma diagnosis (sampling time). This heterogeneous setting may affect the protein profile and as described, these results must be taken as hypothesis-generating and awaits further validation in additional larger and independent cohorts.”

Reviewer 3 Report

Holst et al. aim to investigate whether lymphoma patients with and without concurrent myeloproliferative neoplasia (MPN) can be pathogenetically related. For this purpose, they analyzed formalin fixed, paraffin-embedded (FFPE) primary diagnostic lymphoma specimens from 34 patients using a mass spectrometry-based proteomic approach. In the analysis of lymphoma biopsies from MPN-AITL (n=8) vs R-AITL (n=8) patients authors identified 20 significantly differentially expressed proteins, and in the analysis of lymphoma biopsies from MPN-DLBCL (n=9) vs R-DLBCL (n=9) patients - 34 significantly differentially expressed proteins. Authors used these differences in protein expression for clustering of the lymphomas with and without MPN. Additionally, authors performed immunohistochemical evaluation and quantification of selected proteins, and performed ingenuity pathway analysis on significantly differentially expressed proteins. The study is definitely interesting, novel and original, and the manuscript is clear and concise.

However, the following points need to be considered:

  • It should be clear that differentially expressed proteins of different lymphomas will show separate cluster in PCA. Therefore, it is not very much advisable to make PCA on the differentially expressed proteins, and better to make for example a heatmap on them.
  • Moreover, it is not clear how do the clustering of patients into different groups, meaning that groups are different, strength the hypothesis that the lymphoma and MPN may be biologically related?
  • How do PCA plots look like, if you use all identified proteins?
  • Have you tried to put all four different groups of patients on one PCA plot? Did you see four different clusters? Clustering or overlapping of different groups in this case may strength the hypothesis of biologically relation of overlapping samples.
  • Which peptide was used for the IDH2 identification? May the known mutations in IDH2 gene, described by the authors R172M, R172G and R140Q, influence the digestion of IDH2 protein or change the peptide mass?
  • Use of ingenuity pathway analysis is good to build a hypothesis, but some evaluating experiments should be performed to support these findings.

Author Response

Point-by point response to reviewer comments, Cancers-1429931

Reviewer 3:

Holst et al. aim to investigate whether lymphoma patients with and without concurrent myeloproliferative neoplasia (MPN) can be pathogenetically related. For this purpose, they analyzed formalin fixed, paraffin-embedded (FFPE) primary diagnostic lymphoma specimens from 34 patients using a mass spectrometry-based proteomic approach. In the analysis of lymphoma biopsies from MPN-AITL (n=8) vs R-AITL (n=8) patients authors identified 20 significantly differentially expressed proteins, and in the analysis of lymphoma biopsies from MPN-DLBCL (n=9) vs R-DLBCL (n=9) patients - 34 significantly differentially expressed proteins. Authors used these differences in protein expression for clustering of the lymphomas with and without MPN. Additionally, authors performed immunohistochemical evaluation and quantification of selected proteins, and performed ingenuity pathway analysis on significantly differentially expressed proteins. The study is definitely interesting, novel and original, and the manuscript is clear and concise.

However, the following points need to be considered:

  • It should be clear that differentially expressed proteins of different lymphomas will show separate cluster in PCA. Therefore, it is not very much advisable to make PCA on the differentially expressed proteins, and better to make for example a heatmap on them.

Response: It is true that with the input of only the differentially expressed protein (p<0.05) the likelihood of enabling separation of the two groups are higher than with all identified protein. To visualize this, we have performed the PCA plot with all identified proteins in both analyses and added this in the supplementary information. Furthermore, as the reviewer suggests, a heatmap may present the results in an additional way and more directly show the expression levels. Thus, we have added a heatmap and hierarchical clustering of the data in the supplemental material document (Supplementary Fig S2). In fact, our experience is that it is not always that the groups are separated even when it is only the differentially expressed proteins that are used. If the biological differences between the groups are discrete it may not be possible to separate the groups. In the present case MPN-AITL could be separated from AITL (Fig. 1C). However, MPN-DLCBL could only be partially separated from DLCBL (Fig. 1D). When all identified proteins are used it is not possible to separate the groups as the biological changes apparently are too discrete to be seen.

Please see figure in attached file.

Supplementary Figure S2: Clustering of patients based on significantly differentially expressed proteins and all identified differentially expressed proteins, respectively.

(A) Heatmap and hierarchal clustering with input of the 20 significantly differentially expressed proteins between the MPN-AITL and R-AITL tumors. (B) Heatmap and hierarchal clustering with input of all 1074 identified differentially expressed proteins between the MPN-AITL and R-AITL tumors. (C) PCA with input of all 1074 differentially expressed proteins between the AITL and R-AITL tumors. (D) Heatmap and hierarchal clustering with input of the 34 significantly differentially expressed proteins between the MPN-DLBCL and R-DLBCL tumors. (E) Heatmap and hierarchal clustering with input of all 1141 identified differentially expressed proteins between the MPN-DLBCL and R-DLBCL tumors. (F) PCA with input of all 1141 differentially expressed proteins between the DLBCL and R-DLBCL tumors. 

Abbreviations: AITL, angioimmunoblastic T-cell lymphoma; DLBCL, diffuse large B-cell lymphoma; MPN, myeloproliferative neoplasia; PCA, principal component analysis; R-, reference sample.

  • Moreover, it is not clear how do the clustering of patients into different groups, meaning that groups are different, strength the hypothesis that the lymphoma and MPN may be biologically related?

Response: Surely, we agree with the reviewer in that this is highly hypothetical and to clarify this, we have changed the wordings in the discussion section: “In addition, although hypothetical, our findings are compatible the hypothesis that the lymphomas develop through different pathological mechanisms in that the proteome in the two lymphomas are different, thereby representing two distinct biological diseases. Whereas AITL and DLBCL originates from mature lymphocytes [9,12], it could be postulated that the development of MPN-lymphomas may descend from earlier hematopoietic progenitors harboring genetic alterations that later drive the development of both malignancies. In that case, pathogenetic mechanisms may relate back to early events occurring in the hematopoietic stem cells, and the clonal expansion of cells descending from an abnormal hematopoietic progenitor cell may be responsible for the development of both MPN and lymphoma. Further data supporting this hypothesis are warranted.”

  • How do PCA plots look like, if you use all identified proteins?

Response: See the response above, as expected if the input with all identified proteins is used in the PCA plots the groups are not separated in the PCA analyses (supplementary Fig S2).

  • Have you tried to put all four different groups of patients on one PCAlot? Did you see four different clusters? Clustering or overlapping of different groups in this case may strength the hypothesis of biologically relation of overlapping samples.

Response: We did approach our data in this way initially. However, in that B-cell (DLBCL) and T-cell (AITL) lymphoma are distinct entities and easily recognized and distinguished morphologically, we do not believe that this comparison is relevant in the current setting. To elaborate a bit on this we did expect to see proteins that were differentially expressed both between DLBCL/DLBCL-MPN and AITL/AITL-MPN which could have prompted a hypothesis on common biological differences. However, we did not see this, as stated in the results section. If this is due to the methodology in the study or due to distinct biology in the B- and T- cell lymphoma still awaits further investigation. A PCA plot, see below based on all proteins identified in all samples in all four groups do not show any clustering of the samples, i.e., the samples from each group are spread uniformly in the plot. Again, the biological differences between the groups are apparently too subtle to be visualized based on all proteins. The differences become more apparent when the PCA plot is only based on the differentially expressed proteins.

Please see figure in attached file

  • Which peptide was used for the IDH2 identification? May the known mutations in IDH2 gene, described by the authors R172M, R172G and R140Q, influence the digestion of IDH2 protein or change the peptide mass?

Response: Yes, certainly all three mutations will affect the tryptic digestion since they all involve the disappearance of arginine (R), where trypsin cuts at the C-terminus. Thus, tryptic digestion of a protein with one of the mutations will give one peptide less (since trypsin will not cut at the mutated amino acid). All other peptides from the protein will have the same mass as those occurring from the un-mutated protein. As the protein is identified by searching the tryptic generated peptides against the human database, that does not contain the mutation, it will be all the other unmutated peptides that may be identified. In this analysis we identified 8 unique peptides (NILGGTVFR; DIFQEIFDK; DLAGCIHGLSNVK; DQTDDQVTIDSALATQK; LIDDMVAQVLK; LILPHVDIQLK; LNEHFLNTTDFLDTIK; VCVETVESGAMTK; YFDLGLPNR) in the IDH2 sequence as shown below. None of these involved the mutations. One non-unique peptide (NILGGTVFR) that is localized C-terminal to the R140Q site was also identified. However, this peptide was non-unique and may also occur from IDH1.

  • Use of ingenuity pathway analysis is good to build a hypothesis, but some evaluating experiments should be performed to support these findings.

Response: Ingenuity pathway analysis was performed to get an overview of pathways that could possibly be implicated by the changes in protein compositions between the MPN and non-MPN associated lymphomas. As presented in the study, we find that several pathways in both AITL and DLBCL are linked in various ways, which is a very interesting observation. Thus, the pathway analyses were performed to define a hypothesis-generating starting-point for future more in-depth mechanistic explorations. Future studies must focus on understanding the mechanistic differences in biology between the lymphomas. Such studies must be undertaken as separate studies and were not the scope of the present study. We agree with the reviewer that evaluating experiments to support the findings would be very welcome. However, to accomplish this would require a substantial experimental set-up with a number of approaches with several steps where important decisions should be made, e.g., type of cell line to choose, which protein to knockout or over-express, specific type of experiments to perform, etc. Although this would be interesting and important to understand the pathological mechanisms, the time frame to perform this would be years rather than months and would thus substantially delay the results we already have. We therefore feel this is beyond the scope of the present discovery-based analysis.
